# A Risk Model for Patients with PSA-Only Recurrence (Biochemical Recurrence) Based on PSA and PSMA PET/CT: An Individual Patient Data Meta-Analysis

**DOI:** 10.3390/cancers14215461

**Published:** 2022-11-07

**Authors:** Rie von Eyben, Daniel S. Kapp, Manuela Andrea Hoffmann, Cigdem Soydal, Christian Uprimny, Irene Virgolini, Murat Tuncel, Mathieu Gauthé, Finn E. von Eyben

**Affiliations:** 1Cytel Incorporated, 1050 Winter St, Waltam, MA 02452, USA; rie.voneyben@gmail.com; 2Department of Radiation Oncology, Stanford University School of Medicine, Palo Alto, CA 94305, USA; dskapp@stanford.edu; 3Department of Occupational Health & Safety, Federal Ministry of Defense, 53123 Bonn, Germany; manhoffm@uni-mainz.de; 4Department of Nuclear Medicine, University Medical Center, Johannes Gutenberg University in Mainz, 55101 Mainz, Germany; 5Department of Nuclear Medicine, University of Ankara, Ankara 06100, Turkey; csoydal@yahoo.com; 6Department of Nuclear Medicine, University Hospital in Innsbruck, 6020 Innsbruck, Austria; christian.uprimny@tirol-kliniken.at (C.U.); irene.virgolini@i-med.ac.at (I.V.); 7Department of Nuclear Medicine, Hacettepe University, Ankara 06230, Turkey; murat.tuncel@hacettepe.edu.tr; 8Department of Nuclear Medicine, Incept, Institute Holland, 38100 Grenoble, France; mathieugauthe@yahoo.fr; 9Center of Tobacco Control Research, 5320 Odense, Denmark

**Keywords:** biochemical recurrence, new generation imaging, overall survival, prognostic factors, prostate cancer, restaging, risk models

## Abstract

**Simple Summary:**

We undertook an individual patient data meta-analysis of the overall survival of 1216 patients with PSA-only recurrence of prostate cancer restaged with PSMA PET/CT before salvage treatment. Despite the patients having a low PSA at the recurrence, the restaging PSMA PET/CT markedly predicted the overall survival for the patients with a prescan PSA > 0.5 ng/mL.

**Abstract:**

An individual patient meta-analysis followed 1216 patients with PSA-only recurrence (biochemical recurrence, BCR) restaged with [^68^Ga]Ga-PSMA-11 PET/CT before the salvage treatment for median 3.5 years and analyzed the overall survival (OS). A new risk model included a good risk group with a prescan PSA < 0.5 ng/mL (26%), an intermediate risk group with a prescan PSA > 0.5 ng/mL and a PSMA PET/CT with 1 to 5 positive sites (65%), and a poor risk group with a prescan PSA > 0.5 ng/mL and a PSA PET/CT with > 5 positive sites (9%) (*p* < 0.0001, log rank test). The poor risk group had a five-year OS > 60%. Adding a BCR risk score by the European Association of Urology did not significantly improve the prediction of OS (*p* = 0.64). In conclusion, the restaging PSMA PET/CT markedly predicted the 5-year OS. The new risk model for patients with PSA-only relapse requires a restaging PSMA PET/CT for patients with a prescan PSA > 0.5 ng/mL and has a potential use in new trials aiming to improve the outcome for patients with PSA-only recurrence who have polysites prostate cancer detected on PSMA PET/CT.

## 1. Introduction

Prostate cancer (PCa) is a frequent cancer among men and causes the second highest cancer mortality [1]. It is estimated that worldwide 375,000 men die annually of PCa [1]. Patients who initially present with localized PCa are treated with a curative intent but a quarter to half of the patients develop a recurrence. The first phase of the recurrence is denoted as prostate-specific antigen (PSA)-only recurrence (biochemical recurrence, BCR) because conventional restaging with CT and bone scans is generally negative. Part of the challenge is that half of the men who die initially have had local PCa according to a conventional staging.

In recent years, restaging with prostate-specific membrane antigen (PSMA) PET/CT has an increasingly important role for patients with PSA-only recurrence [2] and guidelines recommend that most patients be evaluated to undergo salvage treatment at a PSA ≤ 0.5 ng/mL. The European Association for Urology (EAU) developed a BCR risk classification that separates patients with PSA-only recurrence in two groups [3].

Complementarily, a study of the patients at Ankara University Hospital reported that the number of positive sites on [^68^Ga]Ga-PSMA-11 PET/CT predicted OS [4]. In a previous research letter on patients with early PSA-only recurrence in an individual patient data (IPD) meta-analysis, we reported that a PSA threshold of 0.5 ng/mL showed two groups that differed in OS [5].

The present report of the IPD meta-analysis cohort aims to elucidate how a prescan PSA, restaging PSMA PET/CT, and the EAU BCR risk classification best predict OS.

## 2. Evidence Acquisition

### 2.1. Selection of Patient Cohorts and Synthesis of Clinical Data

We searched for cohorts of patients with PSA-only recurrence in PubMed using the search words ((Prostate cancer OR prostate neoplasms) AND (prostate specific membrane antigen OR PSMA) AND (positron emission tomography/computed tomography OR PET/CT) AND (overall survival OR OS)). A selection of the hits pointed to nine candidate centers.

The nine centers had evaluated their patients who had PSA-only recurrence and rather low prescan PSA with a restaging PSMA PET/CT. All included patients had follow-up. Included in the present study were five cohorts, as shown in Figure 1. We excluded a center that elected not to provide data on OS, two centers that were unable to provide follow-up on all their patients, and one center that had a median follow-up of less than two years.

Our investigation is a retrospective IDP meta-analysis of all patients with PSA-only recurrence in the cohorts registered at centers in Austria, France, Germany, and Turkey [4,6,7,8,9,10]. The patients had been followed for more than two years and none of the patients should be lost at follow-up.

The included patients were ≥18 years old and had histologically/cytologically proven PCa and no other malignancies. The patients had had local disease at diagnosis and had undergone radical prostatectomy (RP), radiation therapy (RT), or both. They had been followed with regular determinations of PSA, had developed PSA-only recurrence and had undergone a restaging [^68^Ga]Ga-PSMA-11 PET/CT in an early phase of the PSA-only recurrence before they were given salvage treatment.

The patients gave informed consent, and these researchers analyzed and reported deidentified data about the PSA-only recurrence and the follow-up. The Ankara University Medical School approved the protocol for the investigation as of 26 October 2021, reference number 19-607-21.

### 2.2. PSA

The centers measured PSA with sensitive PSA assays. The Ankara [4], Innsbruck [8], and Mainz centers [9] reported the interval between the prescan PSA measurement and the PSMA PET/CT.

### 2.3. PSMA PET/CT

All PSMA PET/CTs were carried out as [^68^Ga]Ga-PSMA-11 PET/CT and followed international guidelines [11]. To optimize the abdominopelvic CT imaging, the centers had given some patients oral or intravenous contrast one to two hours before the PSMA PET/CT. ^68^Ga was given as 1–2 MBq per kg body weight with a median ^68^Ga activity of 189 MBq (range 77–360 MBq). The time to acquisition of the image was 60–100 min.

The centers conducted a diagnostic CT. Acquisition of the PET image followed a standard protocol with imaging of body beds from the skull vertex to the midthigh. Experienced board-certified nuclear medicine physicians and radiologists evaluated and reported the PSMA PET/CT scans.

### 2.4. Definitions

A rising PSA after the initial treatment was defined as PSA rising from unmeasurable PSA levels and increasing in at least two PSA determinations carried out at more than a one-week interval. The center did not use a lower and an upper limit for the prescan PSA.

Androgen deprivation therapy (ADT) was defined as all ADT registered for the cohorts.

A positive site on PSMA PET/CT was defined as a higher mean standard uptake value (SUV_mean_) in the site than in the background according to a previous study [12]. The regional location of positive sites was defined by a molecular imaging TNM classification (miTNM) [13] with the regional stage based on the most advanced positive site on the PSMA PET/CT. In our present study, equivocal lesions were considered as negative.

The PSMA PET/CT findings were defined in three groups giving equal weight to positive sites in the prostate bed and in other locations. “No site” was defined as a negative PSMA PET/CT, “oligosites” was defined as 1–5 positive sites, and “polysites” was defined as >5 positive sites.

The EAU BCR risk classification [3] defines BCR low-risk for patients initially treated with RP as those with a pathologic ISUP grade ≤ 3 and a PSA doubling time > 12 months. The classification defines BCR high-risk for patients treated with RP as those with a pathologic ISUP grade 4 or 5 and a PSA doubling time ≤ 12 months. For patients initially treated with RT, the EAU BCR risk classification defines patients with BCR low-risk as those with a biopsy-derived ISUP grade ≤ 3 and an interval to relapse > 18 months. The classification defines patients initially treated with RT who had BCR high-risk as those with a biopsy-derived ISUP grade 4 and 5 and an interval to relapse ≤ 18 months.

Stereotactic body radiation therapy was abbreviated as SRT. Salvage radiation therapy (SaRT) was defined as the RT for the PSA-only recurrence.

### 2.5. Statistical Analysis

We considered ADT as a confounder and controlled our risk model for patients who had been treated with ADT concomitantly with the initial RP or RT.

We evaluated the prognostic impact of baseline characteristics, findings on restaging PSMA PET/CT, and the EAU BCR risk classification in Cox regression analyses. We evaluated OS using Kaplan–Meier plots, log-rank test, and Cox proportional hazard models. We developed a risk model using the Furnival branch and bound algorith, and compared our new risk model with the EAU BCR risk classification using Harrell’s concordance statistics (c-index).

We adjusted for multiple testing using pairwise comparisons including the Tukey novelty significance difference. To see whether missing data had a significant impact on the statistical analyses, we undertook a sensitivity analysis by a multiple imputation method of fully conditional specification. The Kaplan–Meier plots were truncated at 5 years. We conducted all statistical analyses as two-sided tests and used a *p* value < 0.05 to indicate statistical significance. All statistical analyses were conducted in SAS (SAS Institute Inc., Cary, NC, USA).

## 3. Results

### 3.1. Patients

The meta-analysis includes five cohorts with 1216 patients with PSA-only recurrence, as shown in Figure 1. Clinical characteristics are summarized in Table 1. The management of the patients differed slightly between the centers. The Hacettepe center [6] gave ADT to 48 (50%) of 96 patients as part of the initial treatment. Four centers restaged all patients with PSMA PET/CT, whereas the Paris center [10] carried out the initial restaging with [^18^F]-fluorocholine or [^18^F]-fluciclovine PET/CT and used PSMA PET/CT as a secondary restaging of only the [^18^F]-PET-negative patients.

The Innsbruck center [7] and the Mainz center [9] mainly measured the latest prescan PSA on the day of the restaging PSMA PET/CT (IQR 0–1 days). In contrast in the Ankara center [4] the interval from the latest prescan PSA to the PSMA PET/CT was median 15 days (IQR 11–19 days).

The centers carried out the PSMA PET/CT between 21 June 2014 and 30 April 2020. The latest follow-up of the patients in our analyses was in autumn 2021. The patients were followed-up for a median of 42 months (range 0.2–88 months). During the follow-up 137 (11%) patients died.

### 3.2. PSA and PSMA PET/CT

Each of the five cohorts had an association between the level of the prescan PSA and the number of positive sites on the restaging PSMA PET/CT, as shown in Figure 2.

### 3.3. Cohorts and Restaging PSMA PET/CT

The numbers of positive sites and the regional locations on the restaging PSMA PET/CT in the five cohorts are given in Table 2. On the restaging PSMA PET/CT scans, the five cohorts had similar proportions of patients with no sites, oligosites, and polysites, as shown in Figure 3A–C. A third of the patients had no sites, more than half of the patients had oligosites, and a tenth of patients had polysites.

### 3.4. Treatment after the Restaging PSMA PET/CT

The treatment after the restaging PSMA varied within and between the centers. The Ankara center [4] followed 7 patients with active surveillance and treated 36 patients with SaRT, 31 patients with docetaxel, and 21 patients with abiraterone.

The Hacettepe center [6] followed 23 patients with active surveillance and treated 26 patients with SaRT, 29 patients with SaRT combined with ADT, and 9 patients with docetaxel and ADT, and 3 patients with surgery with or without ADT.

### 3.5. Restaging PSMA PET/CT and OS

The potential prognostic variables were evaluated in a Cox regression analysis, as shown in Table 3. Three centers with a total of 854 patients reported the number of patients who had positive sites in local, regional, and distant regions of the body.

The number of positive sites on the PSMA PET/CT (hazard ratio (HR) = 1.10 per additional positive site, *p* < 0.0001), and regional locations of positive sites (*p* < 0.01) were strongly associated with OS. The number of positive sites had a greater statistical significance on OS than the regional location of the positive sites. For simplification, we lumped the patients into three PSMA PET/CT-defined groups with no sites, oligosites, and polysites.

It was prognostically significant to classify the PSA threshold in two groups and the PSMA PET/CT in three groups. For simplification we developed a risk model with three risk groups: a good risk group with a prescan PSA < 0.5 ng/mL, an intermediate risk group with a prescan PSA > 0.5 ng/mL and one to five positive sites on PSMA PET/CT, and a poor risk group with a prescan PSA > 0.5 ng/mL and more than five positive sites (*p* < 0.0001, log rank test), as shown in Figure 4.

Sensitivity analyses included multiple imputations and showed comparable results for the PSA/PSMA PET risk model with or without imputation for missing data. The *p* value for the new risk model remained <0.0001.

### 3.6. The PSA/PSMA PET Risk Model and the EAU BCR Risk Classsification

The EAU BCR risk classification separated our patients into BCR low- and high-risk groups that differed significantly in OS, as shown Figure 5 (*p* = 0.0001, log rank test). However, our PSA/PSMA PET risk model predicted OS better than the EAU BCR risk classification and had a higher c-index (0.77 vs. 0.68). The prediction of OS in the PSA/PSMA PET/CT risk model did not improve significantly by adding the EAU BCR risk score (*p* = 0.64).

## 4. Discussion

Our multicenter IPD meta-analysis of patients with PSA-only recurrence provided new findings to the previous reports of the study groups [4,5,14]. The five cohorts included had similar correlations between the prescan PSA and the number of sites on PSMA PET/CT and similar proportions of no sites, oligosites, and polysites. The analyses for the new risk score showed that approximately a tenth of the patients with prescan PSA > 0.5 g/mL had polysites PCa on restaging PSMA PET/CT. Although the patients with polysites PCa had a markedly worse OS than the patients with oligosites PCa, the 5-year OS was >60%.

Our study on reported real world data includes the largest group of patients with PSA-only recurrence who had been followed after restaging with PSMA PET/CT. Each added positive site increased the risk of death and our patients with polysites had a higher risk than our patients with oligosites. Thus, the dichotomy between oligosites and polysites did not imply a distinction between two different categories of metastatic PCa.

Adding the EAU BCR risk score did not significantly improve the prediction of 0S by our new risk model. Our patients with polysites PCa demonstrated a much better 5-year OS than was reported overall for patients with de novo stage IV PCa (32%) [15].

Our patients who initially had been treated with RT had positive findings on restaging PSMA PET/CT both if they had had a prescan PSA below or above the Phoenix PSA threshold of 2.0 ng/mL above the nadir PSA after the initial RT [16,17]. The findings for the patients who had had their initial treatment with RT support a restaging PSMA PET/CT even at prescan PSA levels below the Phoenix threshold.

Only 28% of our patients had a prescan PSA < 0.5 ng/mL. The proportion is low relative to a recent goal for patients with PSA-only recurrence that aims to refer more than 90% of the patients with PSA-only recurrence for decisions regarding salvage treatment while the pretreatment PSA is ≤ 0.5 ng/mL [5]. Unfortunately, at present the goal is not achieved. Abghari-Gerst et al. [18] reported that 443 of 2025 (22%) patients with PSA-only recurrence had a prescan PSA of ≤ 0.5 ng/mL.

It is worth noting that Metser et al. [19] reported that none of the patients with PSA-only recurrence had a prescan PSA ≤ 0.5 ng/mL. In contrast, Afshar-Oromieh et al. [20] reported that 856 of 2533 (34%) patients had a prescan PSA ≤ 0.5 ng/mL. Artigas et al. [21] reported that 40% of the patients with PSA-only recurrence had a prescan PSA ≤ 0.5 ng/mL. Zamboglou et al. [22] reported that 500 of 815 (60%) patients with PSA-only recurrence had a prescan PSA ≤ 0.5 ng/mL.

With the growing consensus to restage patients with PSA-only recurrence using PSMA PET/CT, the challenge is how best clinically to implement the new imaging. Regarding the patients with a prescan PSA > 0.5 ng/mL in our study, 65% of the patients had a PSMA PET/CT with one to five positive sites and 9% of the patients had more than five positive sites.

Correspondingly, Metser et al. [19] reported that 18 of 47 (48%) patients with PSA-only recurrence had a prescan PSA > 0.5 ng/mL and polymetastatic PCa on restaging ^18^F-DCFPyL PET/CT. Afshar-Oromieh et al. [20] reported that 1298 of 1548 (83%) had prescan PSA > 0.5 ng/mL and positive sites on PSMA PET/CT. A quarter of these patients (21%) may have had polymetastatic PCa. Artigas et al. [21] reported that 40% of the patients had nonmetastatic PCa on PSMA PET/CT, 43% had oligometastatic PCa and 13% had polymetastatic PCa.

Additionally, other studies supported the prognostic implications of the 0.5 ng/mL threshold for a prescan PSA. Ceci et al. [23] reported that the 3-year event-free survival for patients with PSA-only recurrence and a prescan PSA < 0.5 ng/mL was 70% compared to 40% for patients with a prescan PSA > 0.5 ng/mL. The PSA threshold was significant in multivariate analyses (*p* < 0.001). Wenzel et al. [24] reported that the patients with a negative PSMA PET/CT had a better 5-year metastasis-free survival than the patients with a positive PSMA PET/CT (92% vs. 48%).

Several publications reported that a restaging PSMA PET/CT had impact on the salvage treatment. Kirste et al. [25] reported that patients with PSA-only recurrence and oligometastatic recurrence survived longer free of progression if their treatment included both SRT targeting the positive sites and SaRT to the prostate bed than if the treatment only included the SRT.

Emmett et al. [26] reported that relapsing patients underwent different salvage treatments according to the regional location of positive sites on the restaging PSMA PET/CT. Regarding the 3-year progression-free survival, patients with a negative PSMA PET/CT given SaRT survived better than patients followed with surveillance.

The EAU BCR risk classification also significantly predicted OS for our patients, but our new risk model based on prescan PSA and restaging PSMA PET/CT predicted OS better than the EAU BCR risk classification. The difference may be due to the fact that only our new risk model reflects the tumor burden at the time of the salvage treatment. 

Today many tracers are available for PSMA PET/CT, but our findings with ^68^Ga PSMA PET/CT are relevant also for the other PSMA tracers. For patients with PSA-only recurrence, Hoffmann et al. [27] reported that restaging ^68^Ga-PSMA and ^18^F-PSMA-1007 PET/CT had similar diagnostic performance.

Ongoing trials aims to elucidate the outcome of treatment for patients with PSA-only recurrence restaged with PSMA PET/CT. The PERYTON trial includes patients with a PSA < 1.0 ng/mL and a negative restaging PSMA PET/CT [28]. The trial compares conventional SaRT with hypofractionated SaRT. The ADOPT trial includes patients with a prescan PSA < 10 ng/mL and a restaging PSMA PET/CT that detects oligometastatic PCa [29]. The trial compares SRT targeting the detected metastases with or without ADT.

An additional trial on the optimal treatment of patients with oligometastatic PCa (the LUNAR study (NCT05496959, ClinicalTrials.gov) is exploring the use of ^177^Lutetium PSMA radioligand before SRT targeting the positive sites on PSMA PET/CT. 

An argument against restaging patients with PSA relapse is “lack of evidence” for a “gain from the scans”. However, Meijer et al. [30] reported that patients with PSA-only recurrence who underwent SRT after restaging PSMA PET/CT had a better one-year BCR-free survival than patients who underwent SRT without restaging PET/CT (92% vs. 79%). Our study indicates that it may be better to treat patients with PSA-only recurrence who have polysites PCa at the time of a restaging PSMA PET/CT than later in the clinical course of progressive PCa.

The Advanced Prostate Cancer Consensus Conference (APCCC) experts reported concerns of “under- and overtreatment of patients following restaging with PSMA PET/CT”. The literature indicates that a negative restaging PSMA PET/CT should be considered as false-negative and definitively not as evidence for absence of PCa and need of treatment. PSMA PET/CT most often do not detect lesions with a diameter < 0.5 cm. However, the aggressive salvage treatment of our patients with polysites PCa was relevant and not overtreatment. Ongoing randomized trials evaluate treatments after a restaging PSMA PET/CT. In addition, randomized phase III trials may point out the best treatment of patients with polysites PCa, as indicated in Figure 6.

## 5. Limitations

Our IPD meta-analysis of patients with PSA-only recurrence suffers from many of the limitations of all retrospective studies. The study population was somewhat heterogenous due to the lack of standardized pretreatment selection, technical details of the restaging PSMA PET/CT, and treatment protocols for salvage treatment. The centers carried out the restaging PSMA PET/CT before the Federal Drug Administration (FDA) approved [^18^F] DCFPyL PET/CT for patients with PCa.

## 6. Conclusions

The present study supports the goal that most patients with PSA-only recurrence are given salvage treatment while their pretreatment PSA is < 0.5 ng/mL. Patients with PSA-only recurrence PSA > 0.5 ng/mL, and PSMA PET-detected polysites PCa had a promisingly high (> 60%) 5-year OS.

## Figures and Tables

**Figure 1 cancers-14-05461-f001:**
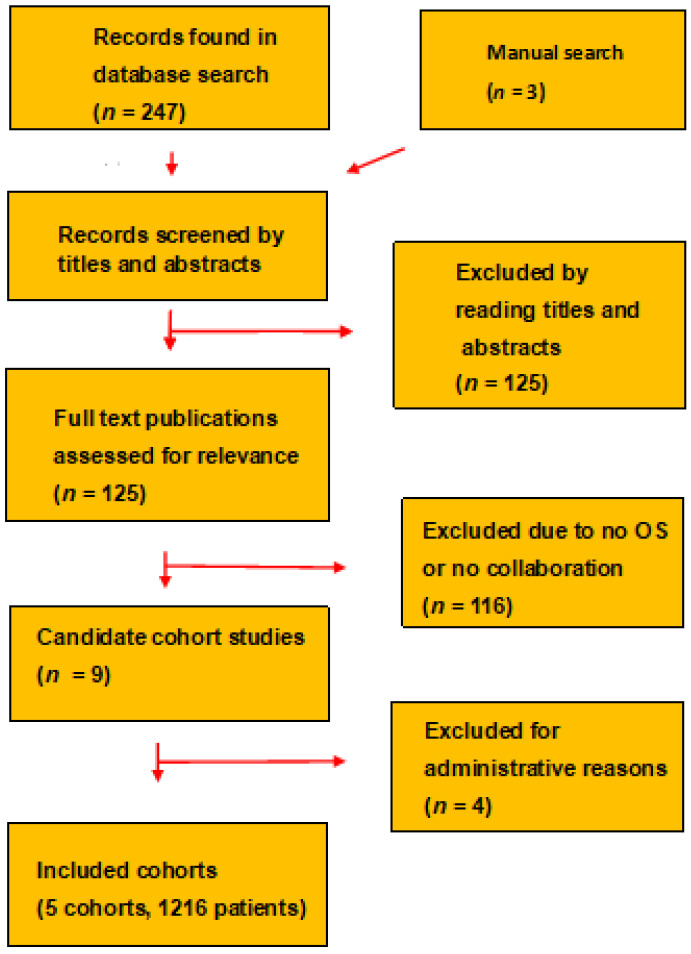
The PRISMA chart diagram shows how the meta-analysis selected and included the cohorts.

**Figure 2 cancers-14-05461-f002:**
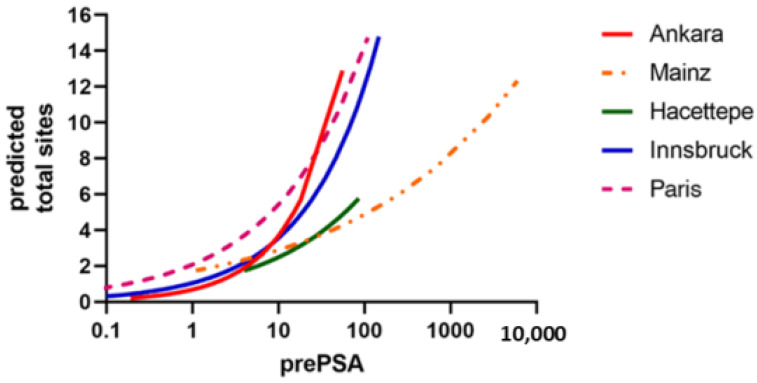
The five cohorts had grossly similar relation between the prescan PSA and the number of positive sites on the restaging PSMA PET/CT. The cohorts are named by the cities for the study centers.

**Figure 3 cancers-14-05461-f003:**
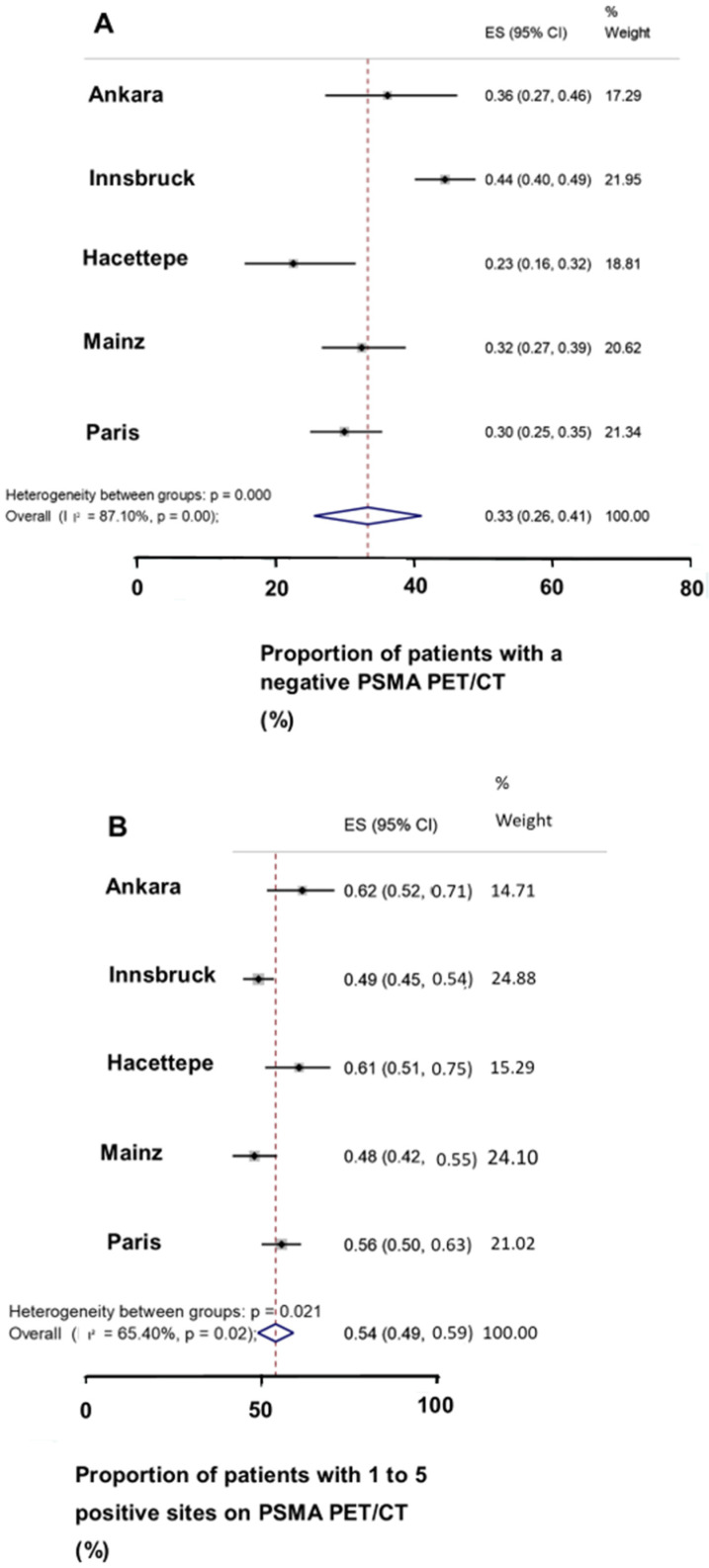
The Forest plot of the patients with PSA-only relapse in the five cohorts. (**A**) a third of the patients had a negative restaging PSMA PET/CT, (**B**) more than half of the patients had one to five positive sites, (**C**) a tenth of the patients had more than five positive sites.

**Figure 4 cancers-14-05461-f004:**
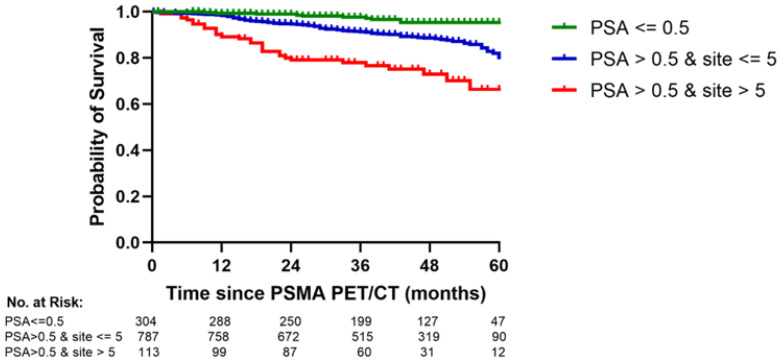
A new risk model shows a significant difference in overall survival between the good-, intermediate-, and poor risk groups (*p* < 0.0001). The figure compares the overall survival for two patient groups by the number of positive findings on restaging PSMA PET/CT: 1–5 sites vs. > 5 sites.

**Figure 5 cancers-14-05461-f005:**
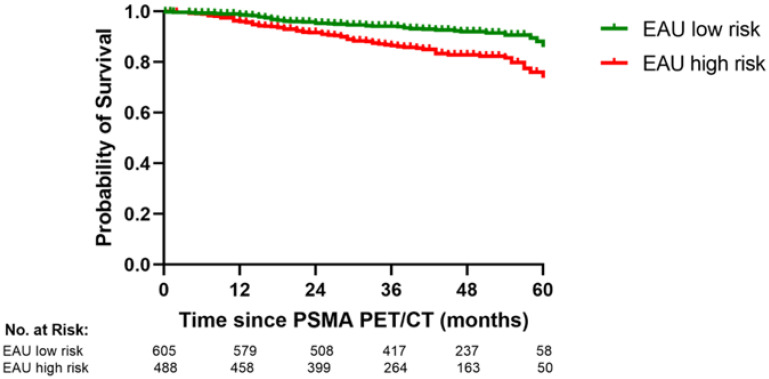
The EAU BCR risk classification significantly separates the patients with PSA relapse into two groups with different overall survival (*p* < 0.0001).

**Figure 6 cancers-14-05461-f006:**
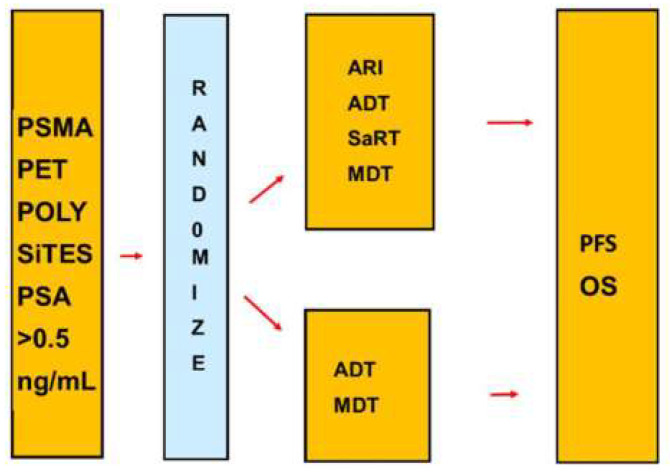
Design for a phase III trial of patients with PSA-only recurrence and high risk according to the new risk model. The trial compares an aggressive multimodality salvage treatment with salvage treatment with established treatments. Abbreviations: ARI second generation androgen receptor inhibitor, MDT metastasis directed therapy, OS overall survival, PFS progression-free survival, SaRT Salvage radiation therapy.

**Table 1 cancers-14-05461-t001:** Clinical characteristics.

Characteristic	Results	
	(Median, IQR, percentage)	Whole range
Age, yrs	68 (62, 73)	41–88
Unknown	118 (10%)	
Time to relapse, months	58 (25, 105)	1–292
Unknown	103 (8%)	
ISUP		
1	123 (10%)	
2	356 (29%)	
3	253 (21%)	
4	158 (13%)	
5	245 (20%)	
Unknown	81 (7%)	
T stage		
1	35 (3%)	
2	389 (32%)	
3	567 (47%)	
Unknown	225 (19%)	
Initial treatment		
RP only	754 (62%)	
RP and RT	145 (12%)	
RT only	304 (25%)	
Unknown	13 (1%)	
Unknown	44 (4%)	
EAU BCR risk score		
Low risk	501 (41%)	
High risk	474 (39%)	
Unknown	241 (20%)	
Follow up time, months	42 (29, 53)	0.2–86
Prescan PSA (ng/mL)	1.4 (0.2, 3.4)	0.1–308.2
Prescan PSA ≤ 0.5 ng/mL	285	
Prescan PSA > 0.5 ng/mL	900	
Unknown	31	

BCR = biochemical recurrence/PSA relapse; EAU = European Association of Urology; ISUP = International Society of Urological Pathology grading of prostate cancer; PSA = serum prostate specific antigen; RP = radical prostatectomy; RT = initial radiation therapy: T = local tumor.

**Table 2 cancers-14-05461-t002:** PSMA PET findings.

Variable	Results (Median, IQR, Percentage)
Total number of regional sites, count	1 (0, 2)
Regional location of positive sites	
No positive sites	369 (30%)
Local sites	187 (15%)
Regional sites	290 (24%)
Distant sites	357 (29%)
Unknown	13 (1%)
EAU BCR risk score	
Insignificant risk	501 (41%)
High risk	474 (39%)
Unknown	241 (20%)
Groupings by no of sites on PSMA PET/CT	
No site	369 (30%)
Oligosites	711 (58%)
Polysites	123 (10%)
Unknown	13 (1%)

Abbreviations as in Table 1.

**Table 3 cancers-14-05461-t003:** Cox model of the individual predictors of overall survival for patients with PSA relapse.

Predictor	HR	95% CI		*p*-Value for the HR	Overall *p*-Value
Age (years)	1	0.98	1.03	0.8263	
Time to relapse (months)	0.99	0.97	0.997	0.0129	
ISUP					0.0146
1 (reference)					
2	1	0.5	2	0.9695	
3	1.2	0.6	2.6	0.6272	
4	1.6	0.8	3.4	0.2028	
5	2.1	1.1	4.2	0.0347	
T stage (1/2 vs. 3)	1.31	0.86	1.99	0.2162	
Initial treatment					0.4739
RP only (reference)					
RP and RT	1.28	0.76	2.15	0.3541	
RT only	1.26	0.82	1.93	0.2994	
Total sites	1.12	1.09	1.15	<0.0001	
Prescan PSA ≤ 0.5 ng/mL (reference)					
Prescan PSA > 0.5 ng/mL	3.706	1.591	8.633	0.0024	
Regional location of positive sites					<0.0001
No sites (reference)					
Oligosites	2.8	1.7	4.6	<0.0001	
Polysites	8.2	4.5	14.8	<0.0001	
No positive sites					
Local sites	3.4	1.8	6.5	<0.0001	
Regional sites	3.5	2	6.3	<0.0001	
Distant sites	4.7	2.7	8.4	<0.0001	
EAU BCR risk score (low vs. high risk)	1.42	0.99	2.06	0.0597	
Groupings by no of sites on PSMA PET/CT					<0.0001

Abbreviations as in Table 1.

## Data Availability

There is no publicly available database for de identified data of the overall cohort.

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
