# Peer review of "A Risk Model for Patients with PSA-Only Recurrence (Biochemical Recurrence) Based on PSA and PSMA PET/CT: An Individual Patient Data Meta-Analysis"

_cancers, 2022, doi:10.3390/cancers14215461_

Round 1
Reviewer 1 Report
In this interesting paper, Authors report a patient-data Meta-analysis regarding PCa biochemical recurrence and risk stratification based on PSA level and PSMA-PET findings. Authors show that this risk stratification better relates to PCa prognosis than EAU risk assessment based on ISUP grading and PSA DT. This work has an important interest, as PSMA-PET is an hot-topic in PCa staging and we still have to incorporate its important utility in clinical practice.
This work present some limitations, as the eterogeneity of study population and the retrospective nature of the study, which need to be underlined by Authors.
Moreover, few minor revisions are required:
-INTRO line 40:"for men" is redundant, Line 41: please reformulate the sentence "each year 400000 men die of prostate cancer": where? from which source you cite this data?
- METHODS: Please explicit the exclusion criteria
Statistical analysis and study results are really well exposed and scientifically sound.
Author Response
Reviewer 1
Introduction delete redundant for men. OK.
Revise and specify 400 000 deaths of PCa per year. Reference 1 estimated 375 000 annual deaths worldwide
Heterogeneity We expanded the Limitations to also stress the heterogeneity between the cohorts.
Reviewer 2 Report
Interesting paper aims to assess the role of PSMA PET in predicting the outcome of patients with biochemical recurrence of Prostate Cancer.
Some comments:
1- an extensive English revision is required
2- the tile should be modified, please introduce biochemical in the replacement of PSA (double mentioned)
3- many errors (references without a specific text, see page 2 line 78)
4-page 3, line 141, 18F what?
5- ADT should be specified only for the first time of appearance
6- The range of PSA> 0.5 ng/mL would be useful (until what maximum value?)
7- the site of polymetastasis should be reported for 2 main reasons a) the identification of the organ and b) for the definition of the subsequent therapy
8- Page 5, line 204-205, it is not clear. The authors reported two separated KM analyses for PSA and 5 lesions and for EAU risk, rather than a combined ones.
9- Page 5, lines 206-211, the sentences are out of the scope of the present study.
10-page 6, lines 262-267, the sentences are out of the scope
11-please add major information about the figure 4 (no unit and no definition of 5)
Author Response
Reviewer 2
1.OK.The revision shows changes on yellow background mainly by Daniel Kapp that has been professor in radiation oncology at Stanford University California for most of his life.
2.OK.The European Association of Urology (EAU) guidelines in 2018 uses the term PSA-only recurrence. So the revision uses the official term. The text stated it previously was called biochemical recurrence, BCR.
3.OK. Sorry, a typo.
4.18F is 18F-PSMA 1007 PET/CT
5.OK. ADT is only spelled out once.
6.No. The centers used no upper limit for PSA at restaging.
7. Table 2 reported the regional location of the positive sites.
8.No. We do not mix the two KM plots into one. A mix would confuse the readers.
9-10. No. We prefer the Discussion debated and remains to debate the topic in a broad perspective.
11.OK, Correct. the number of sites is now stated in the legend to the figure. Good point.
The revision includes further changes, for instance Figure 6.
Round 2
Reviewer 2 Report
The manuscript has improved. Now it is acceptable.